# Design of Microbial Consortia Based on Arbuscular Mycorrhizal Fungi, Yeasts, and Bacteria to Improve the Biochemical, Nutritional, and Physiological Status of Strawberry Plants Growing under Water Deficits

**DOI:** 10.3390/plants13111556

**Published:** 2024-06-04

**Authors:** Urley A. Pérez-Moncada, Christian Santander, Antonieta Ruiz, Catalina Vidal, Cledir Santos, Pablo Cornejo

**Affiliations:** 1Doctorado en Ciencias de Recursos Naturales, Universidad de La Frontera, P.O. Box 54-D, Temuco 4811230, Chile; u.perez01@ufromail.cl; 2Departamento de Ciencias Químicas y Recursos Naturales, Universidad de La Frontera, P.O. Box 54-D, Temuco 4780000, Chile; c.santander01@ufromail.cl (C.S.); maria.ruiz@ufrontera.cl (A.R.); catalina.vidal@ufrontera.cl (C.V.); 3Grupo de Ingeniería Ambiental y Biotecnología, Facultad de Ciencias Ambientales y Centro EULA-Chile, Universidad de Concepción, Concepción 4070411, Chile; 4Centro Regional de Investigación e Innovación para la Sostenibilidad de la Agricultura y los Territorios Rurales, CERES, La Palma, Quillota 2260000, Chile; 5Escuela de Agronomía, Facultad de Ciencias Agronómicas y de los Alimentos, Pontificia Universidad Católica de Valparaíso, Quillota 2260000, Chile

**Keywords:** antioxidant activity, arbuscular mycorrhizal fungi, drought stress, microbial consortia, plant growth-promoting microorganisms, strawberry

## Abstract

Drought affects several plant physiological characteristics such as photosynthesis, carbon metabolism, and chlorophyll content, causing hormonal and nutritional imbalances and reducing nutrient uptake and transport, which inhibit growth and development. The use of bioinoculants based on plant growth-promoting microorganisms such as plant growth-promoting rhizobacteria (PGPR), yeasts, and arbuscular mycorrhizal fungi (AMF) has been proposed as an alternative to help plants tolerate drought. However, most studies have been based on the use of a single type of microorganism, while consortia studies have been scarcely performed. Therefore, the aim of this study was to evaluate different combinations of three PGPR, three AMF, and three yeasts with plant growth-promoting attributes to improve the biochemical, nutritional, and physiological behavior of strawberry plants growing under severe drought. The results showed that the growth and physiological attributes of the non-inoculated plants were significantly reduced by drought. In contrast, plants inoculated with the association of the fungus *Claroideoglomus claroideum*, the yeast *Naganishia albida*, and the rhizobacterium *Burkholderia caledonica* showed a stronger improvement in tolerance to drought. High biomass, relative water content, fruit number, photosynthetic rate, transpiration, stomatal conductance, quantum yield of photosystem II, N concentration, P concentration, K concentration, antioxidant activities, and chlorophyll contents were significantly improved in inoculated plants by up to 16.6%, 12.4%, 81.2%, 80%, 79.4%, 71.0%, 17.8%, 8.3%, 6.6%, 57.3%, 41%, and 22.5%, respectively, compared to stressed non-inoculated plants. Moreover, decreased malondialdehyde levels by up to 32% were registered. Our results demonstrate the feasibility of maximizing the effects of inoculation with beneficial rhizosphere microorganisms based on the prospect of more efficient combinations among different microbial groups, which is of interest to develop bioinoculants oriented to increase the growth of specific plant species in a global scenario of increasing drought stress.

## 1. Introduction

Due to the drastic impact of climate change in recent years, drought has become the limiting factor in crop production worldwide [1]. For instance, in Chile, since around 2010, there has been significant climate variability, with an intensity of dry periods represented by an average precipitation deficit of c.a. 25% [2]. In central and southern Chile, agricultural production is highly dependent on rainfall, and in recent years, a pronounced megadrought has significantly impacted crop production [3].

Strawberries are grown on 389,665 thousand hectares in 73 countries and are among the highest-yielding fruit crops [4,5]. They also boast high levels of antioxidant and phenolic compounds, which are beneficial for human health [6,7]. Despite the large commercial volume of this berry and its nutritional value, strawberries are vulnerable to water stress because of their shallow root system and extensive leaf area, which makes their growth highly susceptible to water deficits [8]. In Chile, strawberry (*Fragaria x ananassa*) cultivation is predominantly practiced in a geographical distribution where the effects of the megadrought have been particularly severe, reaching a water deficit of almost 50% [9].

Water deficits in plants affect several physiological traits such as photosynthesis, carbon metabolism, and chlorophyll content, originating hormonal and nutritional imbalances and noticeably reducing nutrient uptake and transport [10,11]. Furthermore, during water deficits, plants experience significant osmotic stress, which can result in turgor loss, ultimately inhibiting growth and development [1,12]. Reduced relative water content increases the activity of antioxidant enzymes and non-enzymatic antioxidant compounds [13] due to the overproduction of reactive oxygen species and other biochemical features that trigger water deficits in plants [14]. Although the role of antioxidant enzymes and non-enzymatic antioxidant compounds is not limited to improving plant tolerance to drought stress, it also plays an important role in other sources of abiotic stress [15].

In recent years, there has been significant interest in an alternative approach to help plants withstand drought through the use of bioinoculants containing plant growth-promoting microorganisms rhizobacteria (PGPR), arbuscular mycorrhizal fungi (AMF), and yeasts. These microorganisms play a crucial role in enhancing plant tolerance to drought stress by performing various mechanisms, both direct and indirect, to help plants during environmental stress [16,17,18].

Yeasts, a group of unicellular fungi that are typically found in extremely challenging environments, such as those that are very cold, dry, saline, or acidic [19], have received little attention in the context of studying drought tolerance in plants despite their potential to produce indole acetic acid (IAA) and exopolysaccharides (EPS) [16,20]. Previous reports indicate that yeast can also increase the activity of soil enzymes such as protease, urease, glucosidase, phosphatase, and catalase, allowing the plant to account for greater nutrient availability under drought stress conditions [21,22].

Arbuscular mycorrhizal fungi have also been extensively studied regarding their role in enhancing plant tolerance to drought stress. AMF, through their external mycelial network, can efficiently capture and transport water and nutrients to host plants during drought events [23]. This is considered the most significant direct mechanism in the plant-AMF symbiotic relationship [24,25]. The mycorrhizal symbiosis can modify stomata behavior under drought or normal irrigation conditions, playing an important role in plant productivity and also showing a promotion of stomatal conductance (g*_s_*) by up to 24% or more when compared with non-AMF-colonized plants [26].

Although studies on consortia within and between groups of the above-mentioned microorganisms (PGPR, yeasts, and AMF) are scarce, some interactions between AMF and PGPR on plant drought tolerance have also been studied [27,28]. As reported by Ashwin et al. [27], the co-inoculation of *Bradyrhizobium liaoningense* and *Ambispora leptoticha* resulted in improved morphological, physiological, nutritional, and yield variables in three soybean cultivars: MAUS 2, MAUS 212 (drought-susceptible), and DSR 12 (drought-tolerant) when subjected to drought stress [27].

Both PGPR and AMF fungi have been demonstrated to exert a beneficial effect on strawberry plants when subjected to drought stress conditions [29,30,31]. However, despite the growing use of bio-inoculants for enhancing plant tolerance to various stresses, including drought, there is a notable gap in the literature regarding the interaction of yeasts with other microorganisms under drought stress conditions. Additionally, very few studies have assessed the effects of individually inoculated yeasts on plants subjected to drought stress [16,22].

Overall, no systematized work has been carried out to design microbial consortia between PGPB, AMF, and yeasts for use as bioinoculants. This gap in research hinders our understanding of the microbial interactions that can occur in the context of developing bio-inoculants based on microbial consortia [32]. Based on the above, we hypothesized that the combination of different groups of microorganisms with plant growth-promoting traits would improve biomass production, fruit, nutrient uptake, and modulation of antioxidant compounds associated with stress conditions in plants growing under water deficits. Therefore, the aim of this study was to select isolates of AMF, yeasts, and PGPR to generate efficient bioinoculants to improve the biochemical, nutritional, and physiological status of strawberry plants subjected to severe water stress.

## 2. Results

### 2.1. Plant Growth and Arbuscular Mycorrhizal Fungi Traits

Water starvation significantly decreased shoot dry weight (SDW) and root dry weight (RWC) by 37.3% and 17.8%, respectively, in the water-stressed (WS) control compared to the well-watered (WW) control (Table 1). The inoculation of different microbial consortia had a greater effect on SDW than the WS control, where consortia such as Cc+Na+Bc, Cl+Rm+Bt, and Fm+Cg+Bt had an increase by 18% and 16%, respectively (please see the details for microorganism combinations in the footnote of Table 1). On the other hand, WS control and plants inoculated with Fm+Na+Bt showed root dry weights 17.8% lower than the WW control. Moreover, treatments inoculated in consortium with Cc+Na+Bc, Cc+Rm+Pf, Cl+Cg+Bt, and Cl+Cg+Pf increased root dry biomass by 32,3%, 34.3%, 34.3%, and 42.5% over the WS control and 17.6%, 20%, 20%, and 30% over the WW control, respectively. These results are consistent with the root–shoot ratio (Table 1), where plants inoculated with the Cc+Rm+Pf and Cl+Cg+Pf consortia showed the highest values in this ratio with values of 0.9 and 1.0 compared to the WS (0.64) and the WW (0.47) controls.

The results for the RWC in the leaves showed significant differences (*p* ≤ 0.05) among the treatments assessed (Table 1). Uninoculated plants growing under the WS condition reached up to 87.5% RWC, 6.1% less than the WW control and 12.5% less than the plants inoculated with the Cc+Cg+Bc, Cc+Cg+Bt, CC+Cg+Pf, Cc+Na+Bc, Cc+Na+Bt, Cl+Rm+Bc, and Fm+Cg+Bt consortia.

In the WS uninoculated treatment, the fruit number was reduced (Table 1). Strawberry plants inoculated with the consortia Fm+Rm+Bt, Fm+Na+Bc, Cl+Cg+Pf, Cc+Rm+Bt, and Cc+Na+Bc outperformed the WS control in this variable by 83.3%, 81.1%, 82.1%, 82.1%, and 81.1%, respectively, and the WW control by 23.3%, 13.2%, 17.8%, 17.8%, and 13.2%, respectively.

Regarding mycorrhizal colonization frequency (%MCF) and mycorrhization intensity (%MI), all the plants inoculated with AMF showed root formation with the presence of arbuscular mycorrhizal structures. Although MCF values were obtained up to 50%, the MI was reduced with the water deficit, with values below 1.5% (Figure 1). Both MCF and MI were increased in co-inoculation with Fm+Rm+Pf, with values of 50% and 1.2%, respectively.

### 2.2. Photosynthetic Traits

The water deficit strongly and significantly decreased (*p* ≤ 0.01) all the photosynthetic traits studied herein (Figure 2). Photosynthetic rate (A), stomatal conductance (gs), and transpiration (E) were reduced by 90.9%, 77.7%, and 72.3%, respectively, in the WS control compared to the WW control. The highest gs, E, and A values were observed in strawberry plants inoculated with Cc+Na+Bt, Cc+Na+Bc, and Cc+Cg+Bt. On the other hand, the water deficit reduced the quantum yield of photosystem II (ΦPSII) values in the uninoculated WS control, with values of 0.6 compared to the WW control, which presented a value of 0.72 (Figure 2D).

### 2.3. Nutrient Concentration

Nutrient concentration in strawberry plants was affected by water deficits (Table 2). Concentrations of N and P decreased up to 15.9% and 16.6%, respectively, in the WS control compared to the WW uninoculated control. Interestingly, in plants co-inoculated with Cc+Na+Bc, the concentration of nutrients (N, P, and K) increased by 6.6%, 11.7%, and 135.4%, respectively, compared to the WS uninoculated control. Furthermore, the Fm+Cg+Bt consortium tended to have higher P concentrations (1.7 g P kg^−1^) than the WS control (1.5 g P kg^−1^). There were no statistical differences between the WW (7.7 g K kg^−1^) and the WS controls (7.9 g K kg^−1^) for the concentration of K in leaf tissue. Despite this, 59% of the microbial consortia evaluated outperformed the WS and WW uninoculated controls, while the consortia Cc+Na+Bc, Cc+Na+Bt, and Cc+Na+Pf increased the concentration of this nutrient by 135.4%, 112.6%, and 112.6%, respectively, compared to the WS control.

### 2.4. K-Means Clustering Algorithm

An unsupervised k-means clustering algorithm was used to select the treatments with the mean values closest to the WW uninoculated control, which showed the greatest significant differences in the 14 variables evaluated so far (SDW, RDW, root–shoot ratio, RWC, fruit number, MCF, MI, gs, E, A, ΦPSII, N, P, and K).

Two clusters were clearly distinguished (Figure 3). Cluster 1 was composed of the WS uninoculated control and all consortia with similar values. This group was characterized by increased RDW, root–shoot ratio, %MCF, %MI, and fruit number. On the contrary, cluster 2 consisted of the WW uninoculated control and the consortia Cc+Rm+Bc, Cc+Cg+Pf, Cc+Cg+Bt, Cc+Cg+Bc, Cc+Na+Pf, Cc+Na+Bt, Cc+Na+Bc, Cl+Cg+Bt, Cl+Cg+Bc, and Fm+Cg+Bt. This group was characterized by increased SDW, RWC, gs, E, A, ΦPSII, and N, P, and K concentrations. The Fm+Rm+Pf treatment was recovered between clusters 1 and 2.

Most of the consortia inoculated with the fungus *C. claroideum* (Cc) showed the highest values in the variables evaluated, followed by *F. mosseae* (Fm). In this sense and based on the results obtained from the k-means clustering algorithm of all the 14 variables evaluated so far, the consortia Cc+Na+Bc, Cc+Na+Bt, Cc+Rm+Bc, Fm+Cg+Bt, and Fm+Rm+Pf with the values closest to the WW control were selected as the most promising and effective inocula for tolerance to drought in strawberry plants. To determine the two best consortia for future field trials, the above consortia were analyzed for total phenolics; antioxidant activity, such as copper-reducing antioxidant capacity (CUPRAC), the 2,2-diphenyl-1−picrylhydrazyl method (DPPH), and Trolox equivalent antioxidant capacity (TEAC); chlorophyll a, b, and total chlorophyll content; as well as carotenoid and malondialdehyde (MDA) content as a proxy for the biochemical and metabolic improvement by the inoculation.

### 2.5. Total Phenols and Antioxidants

The phenolic compounds were quantified using gallic acid and Trolox as a standard (Figure 4; see Appendix A). No significant statistical differences in total phenolic content and CUPRAC antioxidant activity were found between the five best consortia mentioned above and the WS and WW controls. However, the Cc+Na+Bc and Fm+Cg+Bt consortia increased TEAC antioxidant activity up to 34% and 35.4% more than the WS control. The DPPH antioxidant activity was increased by 30% using the Fm+Rm+Pf consortium and by 22% through the use of Cc+Na+Bc and Fm+Cg+Bt consortia compared to the WS uninoculated control.

### 2.6. Chlorophylls and Carotenoid Contents

Chlorophyll a (Chl a) and total chlorophyll (total Chl) concentrations were higher (Chl a by 19.3% and total Chl by 22.5%, respectively) in strawberry plants inoculated with Cc+Na+Bc than in plants with the WS control (Table 3). In contrast, carotenoid concentrations were lower with Cc+Na+Bc, although no significant differences were obtained between the other consortia and the WS and WW controls (Table 3).

### 2.7. Lipid Peroxidation

In terms of lipid peroxidation, the results showed that the WS uninoculated control produced the highest amount of MDA in strawberries (40.01 μmol g^−1^ FW (fresh weight)) compared to the WW control plants (29.40 μmol g^−1^ FW) (Figure 5). In contrast, inoculation of the consortia Cc+Na+Bc, Cc+Rm+Bc, and Fm+Cg+Bt presented similar values (30.4, 30.4, and 29.9 μmol g^−1^ FW) to the WW control, evidencing a decrease in MDA by 25%, 25%, and 13.25%, respectively, compared to the WS control (Figure 5).

### 2.8. Multivariate Analysis

The 23 measured variables (SDW, RDW, root–shoot ratio, RWC, fruit number, MCF, MI, gs, E, A, ΦPSII, N, P, K, CUPRAC, DPPH, TEAC, Chl a, Chl b, total chlorophyll, CARs, and MDA) of the five best consortia, along with the WS and WW controls without inoculation, were analyzed by means of factorial analysis with principal component (PC) extraction. The two first PCs (PC1 and PC2) accounted for 65.5% of the total experimental variance. Most of the variables were associated in PC1 with the highest variance (46.1%), while a smaller proportion of the variance (19.4%) was obtained in PC2 (Figure 6). In PC1, the variables with a higher contribution were N and P concentrations, SDW, A, Chl a, Chl b, and total Chl. These variables were associated with the consortia Cc+Na+Bc and Cc+Na+Bt and the WW control (Figure 6). On the other hand, CARs were the main negative contributing variable in PC1, associated with the Cc+Rm+Bc consortium and the WS control. Furthermore, in PC2, the variables with the highest positive contribution were gs, A, E, K uptake, and DPPH, and they were associated with the Cc+Na+Bc and Fm+Cg+Bt consortia.

## 3. Discussion

The reduction in water availability resulted in significant changes in the plants, including a decrease in SDW and RDW. However, the effect of co-inoculating AMF, PGPR, and PGPY led to increased biomass production in plants even when subjected to severe drought conditions.

Several studies have demonstrated that co-inoculations of both AMF and PGPR under drought stress conditions result in even greater improvements in biomass production compared to individual inoculations of each of these groups. Begum et al. [28] demonstrated that the co-inoculation of the fungus *Glomus versiforme* and the PGPR *Bacillus methylotrophicus* on tobacco plants resulted in greater increases in dry weight and plant height (11.30 g and 17.99 cm, respectively) compared to treatments where *G. versiforme* (10.18 g and 15.59 cm, respectively) and *B. methylotrophicus* (9.02 g plant^−1^ and 13.95 cm, respectively) were inoculated individually [28].

Ashwin et al. [27] stated that the co-inoculation of *Bradyrhizobium liaoningense* and *Ambispora leptoticha* resulted in increased shoot and root dry weights in three soybean cultivars, MAUS 2, MAUS 212 (drought-susceptible), and DSR 12 (drought-tolerant), under drought stress conditions. According to the authors, these improvements were notably superior to the effects observed in individually inoculated treatments and the single inoculated stressed control [27].

Previous studies have also investigated the interaction between yeast and AMF in maize and faba bean plants under well-watered conditions, revealing a synergistic effect between these microorganisms characterized by enhanced biomass production and nutrient uptake [33,34]. Other studies have explored the interaction between yeast and PGPR in mitigating heavy metal toxicity, such as arsenic [35], and the interplay among yeast, PGPR, and AMF in relation to P content in onion [36]. However, in the latter, the microorganisms were evaluated individually, not in a consortium, and under normal irrigation conditions. In this sense, as far as we know, this is the first study that analyzes the co-inoculation with beneficial microorganisms from three different biological groups in strawberry plants.

Increased root–shoot ratio in plants co-inoculated with *C. claroideum*, *R. mucilaginosa,* and *Pseudomonas frederiksbergensis*, as well as with *C. lamellosum*, *C. guillermondii,* and *P. frederiksbergensis*, suggests that, under water deficit conditions, the predominant allocation of energy was toward root biomass production rather than aboveground biomass. Under drought conditions, several studies suggest that AMF and bacterial inoculation can modify root architecture [27,37]. AMF, with their extensive external mycelial network, can explore a larger volume of soil for water transport [24,25].

Bacteria and yeasts, through the production of auxin-like compounds (AIA), have been shown to increase root biomass under drought stress conditions, contributing to the plant’s ability to tolerate such environmental challenges [16,38] and significantly increasing the water absorptive structures. However, it is important to note that higher root production does not always correlate with increased shoot biomass, as evidenced by our results. The consortia that produced the highest root dry biomass were not necessarily the same as those that yielded the highest shoot dry biomass values. In this sense, the benefits could be evident in more advanced stages of the strawberry crop.

The MCF and MI were increased in most treatments in the consortium. Other studies have found that AMF colonization rates increase when co-inoculated with PGPR [28,39,40], although increases in colonization will depend on the plant species and the microorganisms co-inoculated. Mestre et al. [41] observed that the co-inoculation of the fungus *Rhizophagus irregularis* with the yeast *Tausonia pullulans* in tomato plants delayed mycorrhizal colonization at an early stage, while the co-inoculation with *Saccharomyces eubayanus* slowed colonization throughout the entire evaluation cycle [41]. Conversely, when a commercial tomato cultivar (*Solanum lycopersicum* L. cv. Boludo F1, Monsanto) was co-inoculated with PGPR under drought stress, it resulted in decreased AMF root colonization compared to plants inoculated solely with the fungus [39].

Similar results were obtained in wheat plants under drought conditions, where individual inoculation with a native AMF consortium resulted in a higher MI compared to treatments co-inoculated with bacteria [42]. In our study, among the single inoculation treatments, *F. mosseae* exhibited the highest percentage of MCF and MI, consistent with previous research conducted with this fungus under drought conditions, demonstrating its great capacity to colonize plants under this stress [43,44,45].

On the other hand, in our study, increases in biomass by *C. claroideum*, *N. albida,* and *P. caledonica*, and *F. mosseae*, *M. guillermondii,* and *B. tequilensis* were correlated with increases in N, P, and K concentrations, and the effects of these elements on A, g*_s_*, and E was also evident (Figure 6). The positive effects of N, P, and K concentrations on biomass production and on photosynthetic characteristics support their importance in photosynthetic processes, cell growth, metabolism, and protein synthesis. A decrease in P uptake reduces photosynthetic function, especially the function of RuBisCo and fructose-1,6-bisphosphatase, as well as the high-energy molecules ATP and NADPH [46]. On the other hand, N is a key element in N-containing proteins, enzymes, amino acids, nucleic acids, and plant hormones, as well as the key coenzyme NADPH in CO_2_ fixation in photosynthesis [47]. Moreover, K results are vital for plant osmotic regulation, including the control of stomatal aperture and facilitation of transpiration processes [48]. Here, K was found in higher concentrations in plants inoculated with the consortia Cc+Na+Bc, Cc+Na+Bt, and Cc+Na+Pf. These consortia also showed the highest values of g*_s_*, E, and A. This enabled the plant to maintain its water balance and increase its resistance to drought. Some studies have shown that the inoculation of PGPM, such as AMF and rhizobacteria in plants subjected to water stress, increases K content and, consequently, photosynthesis and RWC [49,50].

In the present study, positive interactions were observed among the fungus *C. claroideum*, the yeast *N. albida*, and the rhizobacterium *B. caledonica* (consortium Cc+Na+Bc), as well as between the fungus *F. mosseae*, the yeast *M. guillermondii*, and the rhizobacterium *B. tequilensis* (Fm+Cg+Bt), facilitating the concentration of nutrients such as N, P, and K into the leaves of the shoot (Table 2). These values were statistically equivalent to the WW control and even surpassed it, particularly evident in K concentrations (Table 2).

Although *C. claroideum*, an AMF species present in the two most effective consortia (Cc+Na+Bc), has received limited attention in the context of plant drought tolerance, the few studies available demonstrate its capacity to enhance various physiological and biochemical traits in plants, enabling them to better withstand drought [18,51]. On the contrary, *F. mosseae* (Fm+Cg+Bt) stands out as one of the most extensively studied AMF species, primarily owing to its remarkable ability to impart drought tolerance to plants. Its beneficial effects have been well-documented in various plant species, including wheat [45,52], trifoliate orange [43], and sesame [44]. The above suggests that the technological application of beneficial microbial consortia can be managed considering the species-specificity among the plant host and the designed (bio)inoculants.

On the other hand, the beneficial effects of the inoculation of rhizobacteria and yeast strains within the two best consortia (Cc+Na+Bc and Fm+Cg+Bt) on the uptake of nutrients such as N could be related to the fact that these groups of microorganisms are synthesizers of ACC deaminase, which degrades ACC into α-ketobutyrate and ammonia, leading to a decrease in ethylene stress [16,53]. This reduction in ethylene increases the plant’s water and nutrient uptake and promotes both shoot and root growth [54,55].

Under drought stress conditions, both yeasts and rhizobacteria have demonstrated the capability to solubilize and mineralize P through the production of various short-chain organic acids and phenolic acids [56]. Additionally, yeasts and rhizobacteria can synthesize and enhance soil enzymatic activity, including phytases, phosphatases, C-P lyases, proteases, ureases, glucosidases, and catalases [22,57], thereby improving the availability of nutrients such as P, Zn, Fe, and ammonia-N [21]. On the other hand, AMF can transport water and nutrients (N, P, and K) released by yeasts and rhizobacteria through their external mycelial network, even under drought conditions [23,28].

Consequently, these tripartite associations involving PGP AMF, rhizobacteria, and yeast can significantly enhance plant tolerance to water deficits induced by drought stress, even surpassing the benefits of individual inoculations of each microorganism. For instance, Silva et al. [58] showed that the co-inoculation of *Rhizophagus clarus* and *Bacillus* sp. improved P uptake 2.4 times more than the individual inoculation of *R. clarus* [58]. Moreover, the co-inoculation of *G. versiforme* and *B. methylotrophicus* in tobacco plants increased the concentrations of N (19.06 mg g^−1^ DW), P (23.93 mg g^−1^ DW), and K (16.84 mg g^−1^ DW) compared to the individual inoculation of *G. versiforme* (17.19, 18.03, 15.47 mg g^−1^ DW) and *B. methylotrophicus* (12.94, 17.39, and 13.81 mg g^−1^ DW) [28]. Similar results were obtained in our study, where inoculation with Cc+Na+Bc and Fm+Cg+Bt increased P (11.3% and 3%, respectively) and N (13.3% and 13.8%, respectively; Table 2) concentrations compared to treatments inoculated individually with each of the AMF (*C. claroideum* and *F. mosseae*).

Antioxidant capacity serves as a crucial measure to evaluate the defense system of a plant; it can be measured by different methods such as CUPRAC, TEAC, and the DPPH method, among others [59]. In this study, TEAC antioxidant activity was higher than DPPH, which suggests that strawberry plants inoculated with Cc+Na+Bc and Fm+Cg+Bt consortia have better antioxidant activity in the presence of anthocyanins compared to hydroxycinnamic acids and flavonols [7,60]. In a study by Parada et al. [6], strawberry plants inoculated with *C. claroideum* and fertilized with different levels of chemical and organic fertilizers showed that TEAC antioxidant activity was increased by AMF [6]. In turn, *F. mossseae* has been shown to increase TEAC and DPPH antioxidant activity under water stress in crops such as wheat [52]. The beneficial effect of PGPM, such as bacteria, yeasts, and AMF, on the increase in non-enzymatic antioxidant activity under water stress conditions has been demonstrated, where the co-inoculation of AMF-bacteria and bacteria-bacteria has been of great interest in recent years [61].

Similar to the findings of Rahimzadeh and Pirzad [62], Azizi et al. [63], Ashwin et al. [64], and Khan et al. [65], in our study, water deficits reduced the concentrations of photosynthetic pigments (Chl a and total chlorophyll). This decrease in chlorophylls under water stress conditions is the result of damage to plant chloroplasts caused by the overproduction of reactive oxygen species [66]. However, plants inoculated with the Cc+Na+Bc consortium were able to increase the total chlorophyll content, which allowed a higher tolerance to stress, a result consistent with the decrease in MDA content in plants inoculated with this consortium.

In the present study, the increase in drought tolerance in strawberry plants was evident even when the treatments were co-inoculated with microbial species belonging to three different groups. This contrasted with the WS non-inoculated treatment and even with the WW one. The above suggests that the use of bio-inoculants composed of a consortium of different microbial groups can optimize plant nutrition and increase biomass production even under severe drought conditions. It also leads to improvements in RWC, gs, E, and A. Additionally, it enhances TEAC and DPPH antioxidant activity, as well as Chl a, b, and total chlorophyll levels, while reducing MDA production.

Future studies should assess plant responses at the molecular level, including transcriptomics and metabolomics, with a special focus on the *C. claroideum*, *N. albida*, and *P. caledonica* (Cc+Na+Bc) consortium. This will provide a deeper understanding of the regulatory mechanisms underlying the effects of tripartite inoculation on drought tolerance. Ultimately, this research can serve as a foundation for designing specific bio-inoculants to noticeably increase the drought tolerance of species-specific crops, optimizing the beneficial functions in the plant rhizosphere.

## 4. Materials and Methods

### 4.1. Microbial Material

#### 4.1.1. Arbuscular Mycorrhizal Fungi (AMF)

Three AMF isolates corresponding to *Funneliformis mosseae* (Fm), *Claroideoglomus lamellosum* (Cl), and *Claroideoglomus claroideum* (Cc) were used. The arbuscular mycorrhizal fungi Fm and Cl were isolated from the hyperarid Atacama Desert, Chile; meanwhile, Cc was isolated from volcanic soils in southern Chile.

#### 4.1.2. Plant Growth-Promoting Rhizobacteria (PGPR)

*Pseudomonas frederiksbergensis* (Pf), *Bacillus tequilensis* (Bt), and *Burkholderia caledonica* (Bc) were used to screen the initial consortia. Bt and Bc bacteria were isolated from the hyperarid Atacama Desert, Chile; meanwhile, Pf was isolated from Rey Jorge Island, Antarctica, Chile.

#### 4.1.3. Plant Growth-Promoting Yeasts (PGPY)

Three PGPY strains corresponding to *Rhodotorula mucilaginosa* (Rm), *Candida guillermondi* (Cg), and *Naganishia albida* (Na) were used to screen the initial consortia. Rm and Cg were isolated from the Piuquenes nonoperational Cu mine tailings, located in the Aconcagua Valley, Los Andes, Valparaíso Region, Chile.

The selection of each microorganism evaluated in this study (AMF, PGPR, and PGPY) was based on the ability of these to enhance plant growth under other environmental stresses in previous trials [16,20,67,68,69].

### 4.2. Location, Plant Material, and Growth Conditions

Seeds of the strawberry cultivar ‘Alexandria’ were obtained from commercial market. Seeds were disinfected with a 2% sodium hypochlorite solution for 3 min, rinsed three times with distilled water, and placed in Petri dishes with sterile filter paper in an incubator at a temperature of 21 °C until germination. Once germinated, they were transferred to 200-cell germination trays using a sterile peat–perlite substrate (7:3 *v*/*v*). After 60 days, the seedlings were transferred to 0.2 L pots in a sterile peat–perlite substrate mixture (7:3 *v*/*v*). The trays and pots were maintained under greenhouse conditions (40–50% relative humidity, 16–23 °C, 16/8 h day/night photoperiod) at the Universidad de La Frontera, Temuco, Chile.

### 4.3. Experimental Design, Treatments, and Inoculation

A fully randomized design was used between three groups of microorganisms (three AMF, three PGPR, and three PGPY) for a total of 27 combinations plus respective controls (water-stressed and well-watered) with 5 experimental units per treatment (N = 145). The 29 treatments used in this study are summarized in Appendix A.

Growth of PGPR and PGPY strains was performed in a 250 mL Erlenmeyer with Luria–Bertani (LB) and Yeast Peptone Dextrose (YPD) broth, respectively, at 28 °C for 24 h and 120 rpm. After incubation, the rhizobacteria and yeast concentration was standardized to an optical density (OD) of 600 nm = 0.6, equivalent to 10^8^ colony-forming units (CFU) mL^−1^ for rhizobacteria and 10^6^ CFU mL^−1^ for yeasts [20]. PGPR and PGPY cells were harvested by centrifugation at 6000× *g* for 10 min, and the final pellet was resuspended in phosphate-buffered saline (PBS) solution.

The AMF were inoculated at the rate of 100 spores g^−1^ of substrate, and the PGPR and PGPY were 1.5 mL of each to obtain 3 mL of the mixed inoculant (first inoculation: 60 days after germination). After 30 days, the strawberry plants were again inoculated with the rhizobacteria and the yeasts (the beginning of the water stress). All mixtures plus controls received 3 mL of a water suspension of AMF inoculum, which was shaken for 1 h and then filtered through qualitative filter paper (90 mm diameter and 180 μm thick). The addition of the filtrate ensured that each treatment contained the native microorganisms from the inoculum but excluded AMF propagules [70].

All 29 treatments were maintained at 85% of water holding capacity (WHC) of substrate (well-watered (WW)) for 30 days after first microbial inoculation. From day 31, irrigation was stopped for all treatments except the WW control (85% WHC) until the substrate reached a moisture level of 30% WHC (severe water stress (WS)). Substrate moisture (30% for water-stress treatments and 85% for well-watered control) was kept and monitored through humidity sensors with Arduino UNO (1.8.19 version) software. Plants were fertilized with Hewitt’s [71] nutrient solution from the second week after inoculation. After 30 days of drought stress, the plants were harvested.

### 4.4. Measurements in Plants

#### 4.4.1. Plant Growth and Arbuscular Mycorrhizal Fungi Traits

Shoot and root dry weights (SDW and RDW, respectively) were determined by drying each organ in an oven at 70 °C for 72 h. The root–shoot ratio was determined after weighing each organ. The relative water content (RWC) was measured according to the method described by Aroca et al. [72]. Briefly, a young leaf from each treatment was weighed for fresh weight (FW) and immediately placed in a vial saturated with water at 5 °C for 48 h. Subsequently, the leaves were reweighed to determine turgor weight (TW) [72]. The samples were dried in an oven at 70 °C for 48 h, and their dry weight (DW) was calculated using the following equation:(1)$$\mathrm{RWC}=\frac{\left(\mathrm{FW}\;-\;\mathrm{DW}\right)}{\left(\mathrm{TW}\;-\;\mathrm{DW}\right)}\times \;100,$$

In addition, 10 days after the onset of water stress, the plants began to flower, and at harvest, some of them showed fruit formation; in this sense, the number of fruits was considered.

Root colonization by AMF was quantified using a dissecting microscope (20–40×) after rinsing a portion of roots in 10% (*w*/*v*) KOH and staining with 0.05% (*w*/*v*) trypan blue in lactic acid [73]. Briefly, roots were covered with a 10% (*w*/*v*) KOH solution and placed in a water bath for 10 min, then washed and covered with a 1 N HCl solution for 10 min at room temperature. After this time, the HCl was discarded, and the roots were covered with 0.05% (*w*/*v*) trypan blue solution and placed in the dark for 24 h. The roots were then washed, and 10 root fragments were selected on a microscope slide for further measurement. Measures of root colonization were taken using the method described by Trouvelot et al. [74], where the mycorrhizal colonization frequency (%MCF), which considers the number of colonized roots in relation to the total number of roots per plant, and mycorrhization intensity (%MI), which indicates the percentage of colonization per root, are quantified by the following equations:(2)$$\mathrm{MCF}=\left(\frac{\mathrm{number}\;\mathrm{of}\;\mathrm{colonized}\;\mathrm{root}\;\mathrm{fragments}\;}{\mathrm{total}\;\mathrm{number}\;\mathrm{of}\;\mathrm{fragments}}\right)\times \;100,$$
(3)$$\mathrm{MI}=\left(\frac{95n5+70n4+30n3+5n2+n1}{N}\right)$$
where n5, n4, n3, n2, and n1 denote the number of fragments of class 5, 4, 3, 2, and 1, and N is the number of root fragments observed in each case.

#### 4.4.2. Nutrient Concentration

Samples were placed in an oven at 70 °C for 72 h, ground in a mortar, and sieved through a 0.5 mm diameter. The quantification of phosphorus (P) was performed by the colorimetric method after the formation of blue molybdate [75]; meanwhile, the nitrogen (N) concentration was obtained by using the Kjeldahl method [75]. Potassium (K) content was determined by atomic absorption spectrophotometry (Unicam SOLAAR, mod. 969, UniCam, Ltd., Cambridge, UK) after acid digestion.

#### 4.4.3. Determination of Total Phenols and Antioxidants

To extract total phenolic compounds and quantify non-enzymatic antioxidant activities, strawberry leaves were crushed and pulverized using liquid nitrogen in a porcelain mortar. Briefly, 0.3 g of ground leaf powder was transferred to 15 mL tubes, and 5 mL of extraction solvent (methanol–formic acid; 95:5, *v*/*v*) was added in the dark with vortexing for 30 s. The samples were sonicated for 60 s at 40% amplitude using a 130 W Sonics & Materials device (Newtown, CT, USA). Subsequently, the samples were agitated for 30 min at 140 rpm in a shaker and then centrifuged at 4000× *g* for 10 min; the supernatants were aspirated using a syringe, passed through a 0.45 μm filter, transferred to a new tube, and stored at −20 °C until analysis. Total phenolic concentrations were determined by the Folin–Ciocalteu method adapted to a microplate spectrophotometer [6]. Briefly, in a micro vial, 15 μL of the extract or standard, 750 μL of deionized water, 75 μL of Folin–Ciocalteu reagent, 300 μL of 20% *w*/*v* Na_2_CO_3_, and 360 μL of deionized water were added. The vials were incubated in darkness for 30 min at 20 °C. Then, 250 μL of the solution was added to the wells of the microplate, and its absorbance was measured at 750 nm in Epoch UV–visible equipment from BioTek (Winooski, VT, USA) using gallic acid solution as standard.

Antioxidant activities such as Trolox equivalent antioxidant capacity (TEAC), cupric reducing antioxidant capacity (CUPRAC), and antioxidant activity of the radical DPPH (2,2-difenil-1-picrilhidrazil) were determined [6]. All determinations were carried out by spectrophotometry adapted to 96-well microplates in Epoch UV–visible equipment from BioTek (Winooski, VT, USA) using Trolox as standard. The results are expressed as Trolox equivalents (TE).

#### 4.4.4. Photosynthetic Traits and Pigments

The photosynthetic rate (A), stomatal conductance (g*_s_*), and transpiration (E) were obtained using the Targas-1 equipment (PP Systems, Amesbury, MA, EE. UU.). The efficiency in photosystem II (PSII) was obtained using the Fluorpen portable equipment (Photon Systems Instruments, Drasov, Czech Republic) and the Fluorpen 1.0 software. Additionally, from the extract obtained in Section 4.4.3, the contents of chlorophyll *a* and *b*, total chlorophyll, and carotenoids were measured [76], and the determinations were made by reading the optical density (OD) using spectrophotometry adapted to 96-well microplates in Epoch UV-visible equipment from BioTek (Winooski, VT, USA) at 645 nm, 663 nm, and 480 nm, respectively.

#### 4.4.5. Lipid Peroxidation

The lipid peroxidation was measured by calculating the amount of malondialdehyde (MDA) produced by the thiobarbituric acid reaction [77]. Lipid peroxides were extracted from 100 mg of leaf tissue previously macerated with liquid nitrogen. Briefly, 100 mg of leaf were weighed into an Eppendorf tube, and 1.5 mL of cold 0.2% trichloroacetic acid (TCA) was added for maceration. It was then centrifuged at 10,000 rpm for 5 min at 4 °C. For chromogen generation, 300 μL of the supernatant were taken and mixed with 1.2 mL of a mixture containing 20% (*w*/*v*) TCA and 0.5% 2-thiobarbituric acid (TBA) and incubated at 95 °C for 30 min. After, the tubes were cooled rapidly in an ice bath and then were centrifuged at 10,000 rpm for 10 min at 4 °C. The supernatants were used for spectrophotometric readings at 440, 532, and 600 nm in a spectrophotometry adapted to 96-well microplates in Epoch UV–visible equipment from BioTek (Winooski, VT, USA).

### 4.5. Data Analysis

One-way ANOVA was performed after the corroboration of the statistical assumptions using Kolmogorov–Smirnov and Levene Tests to check the normality and homogeneity of the data, respectively. Significance in the means of the data was analyzed using the Least Significant Difference (LSD) test at the 0.05 probability level. Additionally, the data were subjected to principal component analysis (PCA) to evaluate the multivariate effect of the established treatments and the relationship between variables. In addition, similar data were identified and grouped using the k-means clustering algorithm. All statistical analyses were performed using R software version 4.3.0.

## 5. Conclusions

For the first time, we have evaluated the use of consortia consisting of three groups of microorganisms, including AMF, bacteria, and yeasts, in strawberry plants growing under drought conditions. The association of the fungus *Claroideoglomus claroideum*, the yeast *Naganishia albida*, and the rhizobacterium *Burkholderia caledonica* (Cc+Na+Bc) significantly promoted the growth of strawberry plants under water deficit stress. Strawberry plants inoculated with this consortium increased biomass production, relative water content, fruit number, net photosynthesis, stomatal conductance, transpiration, nitrogen, phosphorus and potassium concentrations, and chlorophyll *a* and *b* levels, all of which positively influenced the antioxidant system of the plants by reducing malondialdehyde contents.

The Cc+Na+Bc consortium allowed the plants to tolerate a severe water deficit of 30% of water holding capacity. The above results highlight the importance of testing different groups of microorganisms in specific plant species to design efficient consortia and searching for positive results based on the promotion of synergistic and non-detrimental effects. However, further analyses, such as to confirm the effects of inoculants at the omics levels, are needed to deeply understand the attributes displayed for the plant growth-promoting microorganisms and to orient the consortia formulation to be used as bioinoculants in a current scenario where the drought is one of the main limitans to food security worldwide.

## Figures and Tables

**Figure 1 plants-13-01556-f001:**
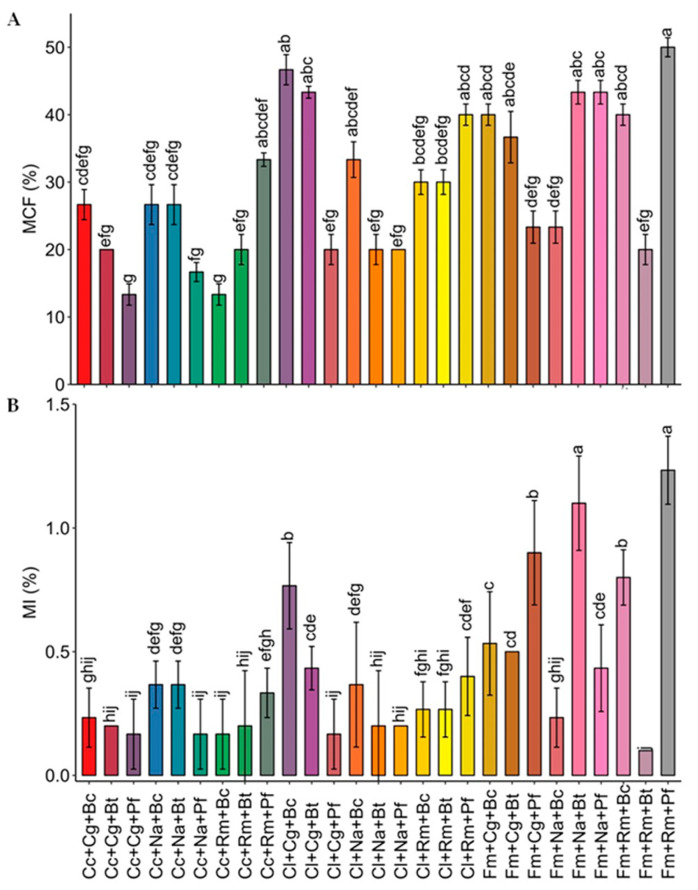
Mycorrhizal colonization frequency (MCF) (**A**) and mycorrhization intensity (MI) (**B**) in strawberry plants inoculated with microbial consortia under water deficit (30% of WHC). The colonization was not registered in the uninoculated controls; therefore, it was excluded from the figure. Cc: *Claroideoglomus claroideum*; Cl: *Claroideoglomus lamellosum*; Fm: *Funneliformis mosseae*; Cg: *Candida guillermondii*; Na: *Naganishia albida*; Rm: *Rhodotorula mucilaginosa*; Pf: *Pseudomonas frederiksbergensis*; Bt: *Bacillus tequilensis*; Bc: *Burkholderia caledonica*. Values represent means ± SE. Different letters indicate significant differences using LSD test (*p* ≤ 0.05).

**Figure 2 plants-13-01556-f002:**
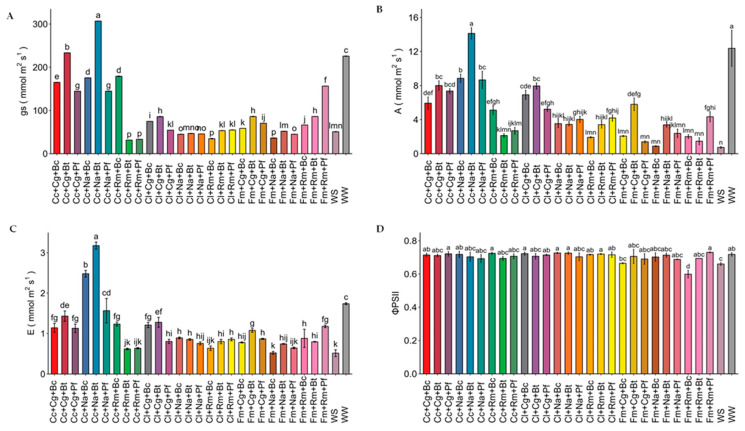
Photosynthetic behavior in strawberry plants inoculated with microbial consortia under water deficit. WW: well-watered (irrigated up to 85% of WHC); WS: water-stressed (irrigated up to 30% of WHC). (**A**) stomatal conductance (gs); (**B**) photosynthesis (A); (**C**) transpiration (E); (**D**) photosystem II (ΦPSII). Cc: *Claroideoglomus claroideum*; Cl: *Claroideoglomus lamellosum*; Fm: *Funneliformis mosseae*; Cg: *Candida guillermondii*; Na: *Naganishia albida*; Rm: *Rhodotorula mucilaginosa*; Pf: *Pseudomonas frederiksbergensis*; Bt: *Bacillus tequilensis*; Bc: *Burkholderia caledonica*. Values represent means ± SE. Different letters indicate significant differences using LSD test (*p* ≤ 0.05).

**Figure 3 plants-13-01556-f003:**
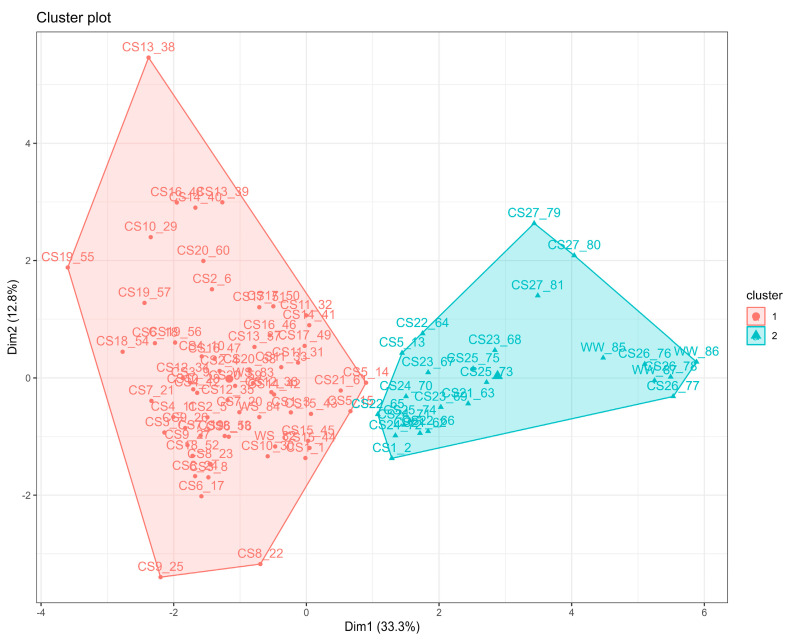
k-means clustering algorithm ordination, clustering similar variables based on an unsupervised machine learning method. WW: well-watered; WS: water-stressed; CS1: Fm+Rm+Pf; CS2: Fm+Rm+Bt; CS3: Fm+Rm+Bc; CS4: Fm+Cg+Pf; CS5: Fm+Cg+Bt; CS6: Fm+Cg+Bc; CS7: Fm+Na+Pf; CS8: Fm+Na+Bt; CS9: Fm+Na+Bc; CS10: Cl+Rm+Pf; CS11: Cl+Rm+Bt; CS12: Cl+Rm+Bc; CS13: Cl+Cg+Pf; CS14: Cl+Cg+Bt; CS15: Cl+Cg+Bc; CS16: Cl+Na+Pf; CS17: Cl+Na+Bt; CS18: Cl+Na+Bc; CS19: Cc+Rm+Pf; CS20: Cc+Rm+Bt; CS21: Cc+Rm+Bc; CS22: Cc+Cg+Pf; CS23: Cc+Cg+Bt; CS24: Cc+Cg+Bc; CS25: Cc+Na+Pf; CS26: Cc+Na+Bt; CS27: Cc+Na+Bc.

**Figure 4 plants-13-01556-f004:**
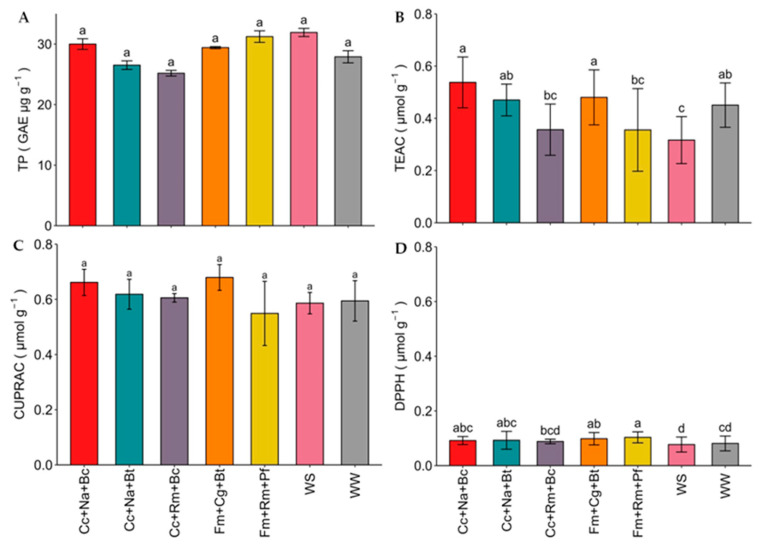
Phenolic compounds and antioxidant activities of leaves of strawberry plants under water stress and inoculated with microbial consortia. (**A**) Total phenols determined by the Folin–Ciocalteu method; (**B**) antioxidant activity (AA) determined by the TEAC (Trolox equivalent antioxidant capacity) method; (**C**) CUPRAC (copper reducing antioxidant capacity) method; (**D**) DPPH (2,2-diphenyl-1-picrylhydrazyl) method. WW: well-watered (irrigated up to 85% WHC); WS: water-stressed (irrigated up to 30% of WHC). Cc: *Claroideoglomus claroideum*; Fm: *Funneliformis mosseae*; Cg: *Candida guillermondii*; Na: *Naganishia albida*; Rm: *Rhodotorula mucilaginosa*; Pf: *Pseudomonas frederiksbergensis*; Bt: *Bacillus tequilensis*; Bc: *Burkholderia caledonica*. Values represent means ± SE. Different letters indicate significant differences using LSD test (*p* ≤ 0.05).

**Figure 5 plants-13-01556-f005:**
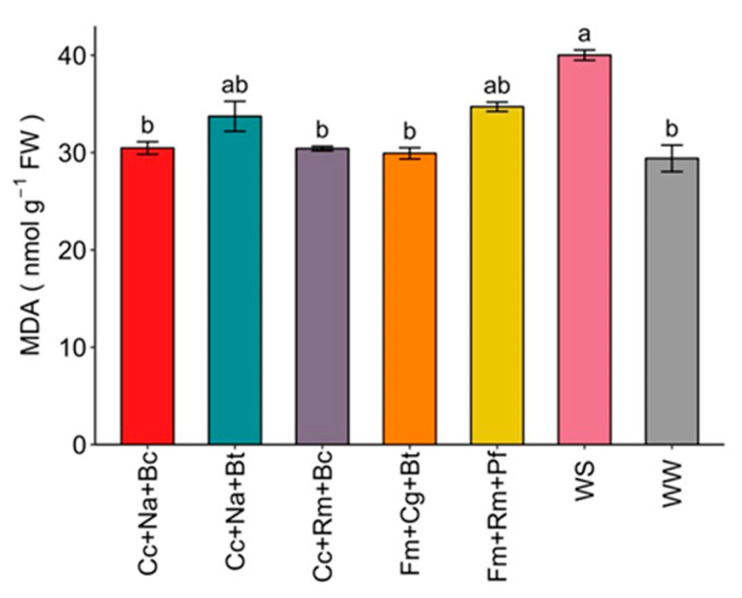
Malondialdehyde (MDA) content in strawberry plant shoots inoculated with microbial consortia under water deficit (irrigated to 30% WHC). WW: well-watered (irrigated up to 85% WHC); WS: water-stressed (irrigated up to 30% of WHC); Cc: *C. claroideum*; Fm: *F. mossea*; Cg: *C. guillermondii*; Na: *N. albida*; Rm: *R. mucilaginosa*; Pf: *P. frederiksbergensis*; Bt: *B. tequilensis*; Bc: *B. caledonica*. Values represent means ± SE. Different letters indicate significant differences using LSD test (*p* ≤ 0.05).

**Figure 6 plants-13-01556-f006:**
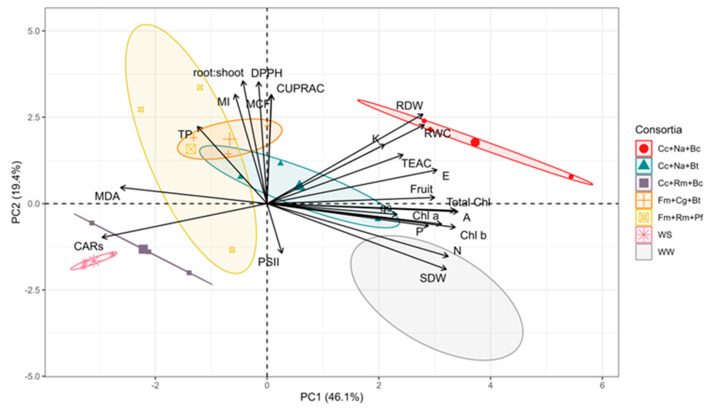
Principal component analysis (PCA) biplot in strawberry plant shoots inoculated with microbial consortia under water deficit (irrigated to 30% substrate) based on biomass production of shoots and root (SDW and RDW); relative water content (RWC); root–shoot relation; mycorrhizal colonization frequency (%MCF); mycorrhization intensity (%MI); net photosynthesis (A); transpiration (E); stomatal conductance (gs); chlorophyll a (Chl *a*); chlorophyll b (Chl *b*); total chlorophyll (Total Chl), carotenoids (CARs); total phenolic compounds (TP); antioxidant activities determined by 2,2-diphenyl-1-picrylhydrazyl (DPPH), Trolox equivalent antioxidant capacity (TEAC), and copper reducing antioxidant capacity (CUPRAC) methods; malondialdehyde (MDA) content; and nitrogen (N), phosphorus (P), and potassium (K) concentrations in the shoot. WS: well-watered (irrigated up to 85% WHC); WS: water-stressed (irrigated up to 30% of WHC).

**Table 1 plants-13-01556-t001:** Shoot (SDW) and root dry weight (RDW), root–shoot ratio, relative water content (RWC), and fruit number in strawberry plants inoculated with microbial consortia and growing under water deficits.

Treatments	SDW (g)	RDW (g)	Root–Shoot	RWC (%)	Fruit Number
Well-watered	5.96 ± 0.09 a	2.82 ± 0.08 d–i	0.47 ± 0.05 h	93.64 ± 0.57 a–e	4.66 ± 0.26 a–d
Water-stressed	3.78 ± 0.02 j	2.36 ± 0.11 i	0.65 ± 0.06 d–h	87.56 ± 0.27 d–g	1.00 ± 1.00 g
Cc+Cg+Bc	3.96± 0.14 e–j	2.36 ± 0.15 hi	0.60 ± 0.02 e–h	100 ± 0.00 a	2.33 ± 1.00 e–g
Cc+Cg+Bt	4.19 ± 0.06 b–h	2.58 ± 0.13 f–i	0.61 ± 0.04 e–h	100 ± 0.00 a	4.66 ± 0.70 a–d
Cc+Cg+Pf	4.34 ± 0.05 b–e	2.54 ± 0.17 f–i	0.59 ± 0.07 e–h	100 ± 0.00 a	3.33 ± 0.83 c–f
Cc+Na+Bc	4.54 ± 0.15 b	3.42 ± 0.05 a–d	0.75 ± 0.08 b–e	100 ± 0.00 a	5.33 ± 1.00 a–c
Cc+Na+Bt	4.3 3± 0.09 b–e	2.44 ± 0.02 g–i	0.56 ± 0.04 f–h	100 ± 0.00 a	1.66 ± 1.18 fg
Cc+Na+Pf	4.37 ± 0.03 b–d	2.43 ± 0.08 g–i	0.56 ± 0.03 gh	96.21 ± 0.66 a–c	4.00 ± 0.50 a–e
Cc+Rm+Bc	4.44 ± 0.22 bc	2.44 ± 0.14 g–i	0.55 ± 0.10 gh	85.53 ± 0.21 e–g	3.66 ± 0.60 b–f
Cc+Rm+Bt	3.90 ± 0.15 f–j	2.95 ± 0.34 b–i	0.75 ± 0.17 b–e	89.16 ± 0.21 b–g	5.66 ± 1.21 ab
Cc+Rm+Pf	3.80 ± 0.04 h–j	3.50 ± 0.18 a–c	0.91 ± 0.07 ab	80.83 ± 0.91 g	4.66 ± 0.26 a–d
Cl+Cg+Bc	3.84 ± 0.04 h–j	2.44 ± 0.03 g–i	0.64 ± 0.00 d–h	90.95 ± 0.04 b–f	4.00 ± 0.86 a–e
Cl+Cg+Bt	4.14 ± 0.07 c–h	3.55 ± 0.37 ab	0.86 ± 0.21 a–c	91.40 ± 0.10 b–f	4.00 ± 0.50 a–e
Cl+Cg+Pf	4.02 ± 0.19 d–j	4.00 ± 0.45 a	1.01 ± 0.30 a	96.63 ± 0.59 ab	5.66 ± 0.64 ab
Cl+Na+Bc	3.70 ± 0.08 ij	2.73 ± 0.31 e–i	0.74 ± 0.17 b–f	91.91 ± 0.88 a–e	4.00 ± 0.00 a–e
Cl+Na+Bt	4.18 ± 0.19 b–h	3.13 ± 0.03 b–f	0.75 ± 0.08 b–e	88.42 ± 0.13 b–g	4.44 ± 0.55 a–e
Cl+Na+Pf	3.88 ± 0.16 g–j	3.31 ± 0.40 b–e	0.85 ± 0.15 a–c	83.43 ± 0.92 fg	3.66 ± 0.30 b–f
Cl+Rm+Bc	3.97 ± 0.10 e–j	2.84 ± 0.21 d–i	0.72 ± 0.09 c–g	100 ± 0.00 a	4.00 ± 0.50 a–e
Cl+Rm+Bt	4.44 ± 0.11 bc	3.30 ± 0.07 b–e	0.74 ± 0.04 b–e	96.12 ± 0.68 a–c	1.66 ± 1.18 fg
Cl+Rm+Pf	4.17 ± 0.03 b–h	3.32 ± 0.43 b–e	0.80 ± 0.21 b–d	95.55 ± 0.78 a–d	2.33 ± 0.37 e–g
Fm+Cg+Bc	4.01 ± 0.09 d–j	2.89 ± 0.22 c–i	0.72 ± 0.14 c–g	86.96 ± 0.72 e–g	3.00 ± 0.56 d–g
Fm+Cg+Bt	4.46 ± 0.11 bc	2.93 ± 0.09 b–i	0.66 ± 0.02 d–g	100 ± 0.00 a	4.00 ± 0.00 a–e
Fm+Cg+Pf	4.07 ± 0.02 c–i	3.04 ± 0.03 b–g	0.75 ± 0.01 b–e	91.49 ± 0.87 b–f	4.00 ± 0.00 a–e
Fm+Na+Bc	4.46 ± 0.15 bc	2.58 ± 0.05 f–i	0.58 ± 0.02 e–h	87.80 ± 0.34 c–g	5.33 ± 0.25 a–c
Fm+Na+Bt	4.27 ± 0.20 b–g	2.36 ± 0.11 i	0.56 ± 0.11 gh	96.33 ± 0.64 ab	4.00 ± 0.50 a–e
Fm+Na+Pf	3.94 ± 0.09 e–j	2.75 ± 0.09 e–i	0.70 ± 0.07 c–g	90.51 ± 0.93 b–f	3.66 ± 0.79 b–f
Fm+Rm+Bc	4.33 ± 0.04 b–e	2.99 ± 0.24 b–h	0.69 ± 0.10 c–g	85.85 ± 0.44 e–g	3.33 ± 0.83 c–f
Fm+Rm+Bt	4.21 ± 0.05 b–h	3.14 ± 0.19 b–f	0.75 ± 0.10 b–e	85.24 ± 0.17 e–g	6.00 ± 1.08 a
Fm+Rm+Pf	4.29 ± 0.13 b–f	2.80 ± 0.17 d–i	0.66 ± 0.11 d–g	92.00 ± 0.83 a–e	4.00 ± 0.86 a–d

Well-watered: irrigated up to 85% water holding capacity (WHC); water-stressed: irrigated up to 30% WHC. Cc: *C. claroideum*; Cl: *C. lamellosum*; Fm: *F. mossea*; Cg: *C. guillermondii*; Na: *N. albida*; Rm: *R. mucilaginosa*; Pf: *P. frederiksbergensis*; Bt: *B. tequilensis*; Bc: *B. caledonica*. Values represent means ± SE. Different letters indicate significant differences using LSD test (*p* ≤ 0.05).

**Table 2 plants-13-01556-t002:** Nitrogen (N), phosphorus (P), and potassium (K) concentrations in strawberry plant shoots after inoculation with microbial consortia under water deficit (irrigated to 30% water holding capacity (WHC)).

Treatments	N	P	K
	(g kg^−1^ Dry Matter)		
Well-watered	13.07 ± 0.80 a	1.84 ± 0.02 a	7.77 ± 0.29 ij
Water-stressed	11.33 ± 0.23 bc	1.51 ± 0.02 c–g	7.96 ± 0.13 ij
Cc+Cg+Bc	9.80 ± 0.70 d–h	1.59 ± 0.00 b–e	15.42 ± 0.32 bc
Cc+Cg+Bt	9.80 ± 0.70 d–h	1.50 ± 0.12 c–h	14.72 ± 0.88 b–d
Cc+Cg+Pf	9.57 ± 0.40 d–h	1.49 ± 0.07 c–h	13.30 ± 0.51 c–f
Cc+Na+Bc	12.13 ± 0.40 ab	1.75 ± 0.10 ab	18.65 ± 0.19 a
Cc+Na+Bt	10.73 ± 0.40 b–e	1.65 ± 0.17 a–d	16.87 ± 0.19 ab
Cc+Na+Pf	9.80 ± 0.00 d–h	1.56 ± 0.07 b–f	16.87 ± 0.19 ab
Cc+Rm+Bc	10.27 ± 0.40 c–g	1.63 ± 0.12 a–d	11.61 ± 1.38 d–h
Cc+Rm+Bt	9.80 ± 0.00 c–h	1.55 ± 0.15 b–f	12.98 ± 0.32 c–g
Cc+Rm+Pf	8.87 ± 0.80 gh	1.49 ± 0.05 c–h	6.85 ± 2.11 ij
Cl+Cg+Bc	9.45 ± 0.35 e–h	1.60 ± 0.12 abc	11.61 ± 0.26 d–h
Cl+Cg+Bt	8.87 ± 0.40 gh	1.53 ± 0.08 b–g	9.02 ± 0.52 h–j
Cl+Cg+Pf	9.10 ± 0.00 fgh	1.35 ± 0.05 f–i	12.35 ± 0.08 c–g
Cl+Na+Bc	9.67 ± 0.20 d–h	1.27 ± 0.18 hi	2.38 ± 0.27 k
Cl+Na+Bt	9.80 ± 0.00 d–h	1.59 ± 0.08 b–e	14.38 ± 0.35 b–e
Cl+Na+Pf	10.03 ± 0.40 c–h	1.49 ± 0.11 c–h	11.27 ± 0.85 e–h
Cl+Rm+Bc	8.75 ± 0.35 h	1.55 ± 0.10 b–g	11.18 ± 0.65 f–h
Cl+Rm+Bt	10.50 ± 0.70 c–f	1.53 ± 0.27 b–g	12.82 ± 0.45 c–g
Cl+Rm+Pf	8.67 ± 0.46 h	1.39 ± 0.21 e–i	6.80 ± 0.33 j
Fm+Cg+Bc	9.80 ± 0.00 d–h	1.51 ± 0.05 c–g	7.47 ± 0.23 ij
Fm+Cg+Bt	10.97 ± 0.40 bcd	1.72 ± 0.05 abc	12.20 ± 0.39 d–g
Fm+Cg+Pf	8.97 ± 0.20 gh	1.39 ± 0.03 e–i	6.08 ± 0.18 j
Fm+Na+Bc	9.10 ± 0.70 fgh	1.31 ± 0.15 ghi	6.56 ± 0.35 j
Fm+Na+Bt	9.33 ± 0.40 e–h	1.18 ± 0.09 i	6.56 ± 0.57 j
Fm+Na+Pf	9.10 ± 0.00 fgh	1.45 ± 0.12 d–h	9.96 ± 0.65 g–i
Fm+Rm+Bc	9.10 ± 0.00 fgh	1.53 ± 0.08 b–g	6.02 ± 0.08 j
Fm+Rm+Bt	10.03 ± 0.80 c–h	1.45 ± 0.07 d–h	6.08 ± 0.59 j
Fm+Rm+Pf	10.97 ± 0.40 bcd	1.63 ± 0.12 a–d	6.50 ± 0.24 j

Well-watered: irrigated up to 85% WHC; water-stressed: irrigated up to 30% WHC. Cc: Claroideoglomus claroideum; Cl: Claroideoglomus lamellosum; Fm: Funneliformis mosseae; Cg: Candida guillermondii; Na: Naganishia albida; Rm: Rhodotorula mucilaginosa; Pf: Pseudomonas frederiksbergensis; Bt: Bacillus tequilensis; Bc: Burkholderia caledonica. Values represent means ± SE. Different letters indicate significant differences using LSD test (*p* ≤ 0.05).

**Table 3 plants-13-01556-t003:** Effect of microbial consortia on chlorophylls and carotenoid content in strawberry plants under drought stress conditions.

Treatments		Chl *a* (mg g^−1^ FW)	Chl *b* (mg g^−1^ FW)	Total Chl (mg g^−1^ FW)	CARs (mg g^−1^ FW)
Water Condition					
Control (85% WHC)	Watered	0.53 ± 0.04 ab	0.58 ± 0.10 ab	1.15 ± 0.08 b	0.15 ± 0.03 a
Control (30% WHC)	Stressed	0.46 ± 0.01 bc	0.54 ± 0.01 abc	1.00 ± 0.02 c	0.16 ± 0.01 a
30% WHC	Cc+Na+Bc	0.57 ± 0.14 a	0.62 ± 0.10 a	1.29 ± 0.08 a	0.12 ± 0.03 b
	Cc+Na+Bt	0.36 ± 0.04 de	0.52 ± 0.04 bcd	0.85 ± 0.03 de	0.15 ± 0.02 a
	Cc+Rm+Bc	0.26 ± 0.05 f	0.43 ± 0.00 d	0.73 ± 0.01 f	0.15 ± 0.02 a
	Fm+Cg+Bt	0.33 ± 0.05 ef	0.48 ± 0.05 cd	0.82 ± 0.07 ef	0.16 ± 0.04 a
	Fm+Rm+Pf	0.43 ± 0.07 cd	0.47 ± 0.05 cd	0.93 ± 0.02 cd	0.15 ± 0.01 a

Chl *a*: chlorophyll *a*; Chl *b*: chlorophyll *b*; Total Chl: total chlorophyll; CARs: Carotenoids; FW: fresh weight. Cc: *Claroideoglomus claroideum;* Fm: *Funneliformis mosseae*; Cg: *Candida guillermondii;* Na: *Naganishia albida;* Rm: *Rhodotorula mucilaginosa;* Pf: *Pseudomonas frederiksbergensis;* Bt: *Bacillus tequilensis;* Bc: *Burkholderia caledonica.* Values represent means ± SE. Different letters indicate significant differences using LSD test (*p* ≤ 0.05).

## Data Availability

Data are contained within the article and Appendix A.

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
