# Peer review of "Design of Microbial Consortia Based on Arbuscular Mycorrhizal Fungi, Yeasts, and Bacteria to Improve the Biochemical, Nutritional, and Physiological Status of Strawberry Plants Growing under Water Deficits"

_plants, 2024, doi:10.3390/plants13111556_

Round 1

Reviewer 1 Report

Comments and Suggestions for Authors

The manuscript deals with the strawberry response to drought and the assessment of microorganisms in mitigating biochemical and physiological effects of drought. The study is interesting, but it has some flaws. The paper requires English revision. Introduction and Discussion should be reorganized. The details are listed below:

Abstract:

L22-24: rephrase

L32-35: add some % changes of examined parameters between treatments

L38, 129: what the Authors mean by ‘a la carte’ in the context of bioinoculants, and in other parts of the text

Introduction:

L59-64: emphasize the ubiquitous role of antioxidant enzymes in mitigating different abiotic stresses (pesticides, drought, etc.). Refer to https://doi.org/10.1016/j.chemosphere.2022.136284

L66-69: remove

L77-81: remove, it was not examined in the study

L71-120: shorten these paragraphs

Results:

L176: indicate full names of these parameters

L229-230: Latin names in italics

L270: lipid peroxidation

L307: ‘oxidative damage to lipids’ – rephrase

Discussion:

Shorten the Discussion. Compare the results of photosynthetic pigments with other studies. Compare the status of antioxidants of this study in the context of beneficial microorganisms  with plants inoculated by plant pathogens. For example wheat inoculated with F. culmorum had higher activity of CAT and POD, but lower SOD activity (refer to https://doi.org/10.3390/agronomy13051378)

Materials and Methods:

L476: rephrase

L484: seeds or seedlings?

L491-492: ‘at the Departamento …’ - remove

L514-520: when plants were collected for analysis?

L548: explain SDW

L566-568: briefly describe

L582: lipid peroxidation

Comments on the Quality of English Language

Moderate editing of English language required

Author Response

Abstract:

L22-24: rephrase

R: Done. From L24 to L25 has been rephrased.

L32-35: add some % changes of examined parameters between treatments

R: Done. Percentage changes of the examined parameters between treatments were added. Please, see L35-L36.

L38, 129: what the Authors mean by ‘a la carte’ in the context of bioinoculants, and in other parts of the text

R: In our current research we are searching for to design specific bioinoculants for specific plant species. Here we are showing the results only for strawberry; therefore, to avoid confusion to the reader, the term ‘a la carte’ has been removed throughout the text.

Introduction:

L59-64: emphasize the ubiquitous role of antioxidant enzymes in mitigating different abiotic stresses (pesticides, drought, etc.). Refer to https://doi.org/10.1016/j.chemosphere.2022.136284

R: Thank you for your suggestion. The ubiquitous role of antioxidant enzymes and non-enzymatic antioxidant compounds in the mitigation of various abiotic stresses was highlighted. Please, see L64-L69.

L66-69: remove

R: Done.

L77-81: remove, it was not examined in the study

R: We agree. Done.

L71-120: shorten these paragraphs

R: Done. Some paragraphs were removed from the introduction.

Results:

L176: indicate full names of these parameters

Response: these lines correspond to Figure 1 and microorganisms have full names.

L229-230: Latin names in italics

Response: the correction was made (L221-222)

L270: lipid peroxidation

R: Done.

L307: ‘oxidative damage to lipids’ – rephrase

R: Unified throughout the text by malondialdehyde (MDA) content. Thank you for your suggestion.

Discussion:

Shorten the Discussion. Compare the results of photosynthetic pigments with other studies. Compare the status of antioxidants of this study in the context of beneficial microorganisms  with plants inoculated by plant pathogens. For example wheat inoculated with F. culmorum had higher activity of CAT and POD, but lower SOD activity (refer to https://doi.org/10.3390/agronomy13051378)

R: Done, the discussion was shortened. Additionally, during the discussion, other studies were presented to compare the results of photosynthetic pigments (L456-463). On the other hand, we are very grateful for your appreciation in the discussion regarding the results of this study, but within the parameters evaluated, the antioxidant enzyme activity (CAT, POD and CAT) was not taken into account, only the antioxidant activity of non-enzymatic compounds (total phenols, CUPRAC, TEAC and DPPH activity). Therefore, we consider that there are other more related studies to be mentioned. However, we will take it into account in a forthcoming publication from our group, where this antioxidant enzyme activity has been evaluated.

Materials and Methods:

L476: rephrase

R: L472-493. Done, please, see the new manuscript version.

L484: seeds or seedlings?

R: The changes have increased the number of queues. Seedlings were exchanged for seeds in L495

L491-492: ‘at the Departamento …’ - remove

R: at the Departamento …. was removed in L502.

L514-520: when plants were collected for analysis?

R: in L531 was mentioned the time when the plants were harvested.

L548: explain SDW

R: SDW is the abbreviation for shoot dry weight. But in L566, the formula has been removed. The nutrient results are expressed in g kg-1 dry matter (Table 2).

L566-568: briefly describe

R: In L548-552 the method was described briefly

L582: lipid peroxidation

R: has been changed, as previously mentioned

Thank you for your comments and the time to review our manuscript.

Reviewer 2 Report

Comments and Suggestions for Authors

For the first time, the authors evaluated the use of consortia consisting of three groups of microorganisms, including AMF, bacteria, and yeasts, in strawberry plants growing under drought condition.

The association of the AMF fungus Claroideoglomus claroideum, yeast Naganishia albida, and rhizobacterium Burkholderia caledonica (Cc+Na+Bc) significantly promoted the growth of strawberry plants under water deficit stress. Strawberry  increasING biomass production.

The topic is really interesting as f strawberry plants are cultivated and consumed worldwide. I highligth the economic importance of this plant, and thus its improvedcultivation. Also the importance of antioxidants present in this crop , considered red fruit, must be highlighted.

Author Response

For the first time, the authors evaluated the use of consortia consisting of three groups of microorganisms, including AMF, bacteria, and yeasts, in strawberry plants growing under drought condition.

The association of the AMF fungus Claroideoglomus claroideum, yeast Naganishia albida, and rhizobacterium Burkholderia caledonica (Cc+Na+Bc) significantly promoted the growth of strawberry plants under water deficit stress. Strawberry increasing biomass production.

The topic is really interesting as f strawberry plants are cultivated and consumed worldwide. I highligth the economic importance of this plant, and thus its improved cultivation. Also the importance of antioxidants present in this crop, considered red fruit, must be highlighted.

R: Thank you for the positive global evaluation of our manuscript. We are pleased since our complete and hard work was well understood and in your opinion is a good and interesting scientific contribution including the food and health implications.

Round 2

Reviewer 1 Report

Comments and Suggestions for Authors

The manuscript has been improved. I have no more comments.